biomechanics/behaviour

aerodynamics, kinematics, animal flight

**Author for correspondence:**
Per Henningsson
e-mail: per.henningsson@biol.lu.se

# Flying through gaps: how does a bird deal with the problem and what costs are there?

## Per Henningsson

Department of Biology, Lund University, Ecology Building, Sölvegatan 35, 223 62 Lund, Sweden

PH, 0000-0003-2640-1067

Animals flying in the wild often show remarkable abilities to negotiate obstacles and narrow openings in complex environments. Impressive as these abilities are, this must result in costs in terms of impaired flight performance. In this study, I used a budgerigar as a model for studying these costs. The bird was filmed in stereo when flying through a wide range of gap widths from well above wingspan down to a mere 1/4 of wingspan. Three-dimensional flight trajectories were acquired and speed, wingbeat frequency and accelerations/ decelerations were calculated. The bird used two different wing postures to get through the gaps and could use very small safety margins (down to 6 mm on either side) but preferred to use larger when gap width allowed. When gaps were smaller than wingspan, flight speed was reduced with reducing gap width down to half for the smallest and wingbeat frequency was increased. I conclude that flying through gaps potentially comes with multiple types of cost to a bird of which the main may be: (i) reduced flight speed increases the flight duration and hence the energy consumption to get from point *A* to *B*, (ii) the underlying U-shaped speed to power relationship means further cost from reduced flight speed, and associated with it (iii) elevated wingbeat frequency includes a third direct cost.

## 1. Introduction

Flying animals have an astonishing ability to negotiate obstacles and narrow openings when flying in complex environments such as forest habitats. I have many times watched a bird elegantly fly through a dense forest at high speeds without touching any of the branches and in awe wondered: how do they do it? But I have also wondered what it costs for the animals to fly in such an environment. How much is their normal flight performance, as

when flying in an open habitat, impaired by having to fly in a complex and cluttered one? Previous work on birds flying past obstacles and through narrow openings have focused on several different aspects including the visual processing of obstacles along the flight path [1,2], the ability of the bird to assess the width of the gap in relation to its own wingspan [3,4], the bird's ability to make split-second decision which gap to use if presented with two different widths [5] or what route to choose through an obstacle course [6]. These studies have revealed a lot about how the birds are indeed capable of flying through complex environments.

For this study, I have focused on the flight performance of a bird flying through various gaps including very narrow ones, exploring the limits particularly with focus on the types of cost that may be associated with it in terms of changes in flight speed and kinematics that the bird is forced to do when negotiating the gap. My aim has been to arrive at an understanding of a concept of what costs may be involved in flying through gaps and not to quantify these costs for any particular species. To study this, I have chosen to use a tame female budgerigar (*Melopsittacus undulates*) as a model. Budgerigars are excellent flyers and are easy to train, particularly if they are already tame and they have been used many times for various studies on flight over the years [3,5,7–12], including flight through gaps ([3–5] and summarized in [13]). In these previous studies on budgerigars flying through gaps, the overall behaviour of the budgerigars have been thoroughly described and established using several individuals. Therefore, in this study, I have focused on a single individual flying through a wide range of gap widths, from well above wingspan down to 1/4 of the wingspan and done an in-depth analysis of the three-dimensional flight trajectories. I have examined the effect of the gap width on flight performance and behaviour: flight speeds, accelerations and decelerations, wingbeat frequency, braking behaviour, safety margins and altered kinematics. I have viewed these aspects both in the perspective of trying to understand how a bird deals with the challenge of flying though a gap and in the perspective of what it costs for the animal to do so. The costs associated are put in the perspective of animals flying in complex and cluttered environments in the wild. The solutions that the bird uses may inspire engineers that are trying to develop small agile flying devices that can handle complex environments.

## 2. Methods

### 2.1. The budgerigar

For this study, I chose to use a tame female budgerigar (a family pet) of cobalt-blue English type. The bird is allowed to fly freely inside the house large portions of the day ensuring that it is kept fit. It is very used to both me and the second bird handler (my wife), which facilitated the experiments. The bird was flown during experiments in our home, meaning the bird was comfortable and calm in its familiar surroundings during our experiments which I believe is key to succeeding in getting a bird to perform these advanced flights repeatedly on demand. Required training of the bird was a minimum, we flew the bird a few times through all panel widths prior to starting the experiments, but the bird was already able to perform the flights at the very start and the behaviour of flying between the two handlers came naturally without effort. The bird was weighed just before or after each flight session and wing area was measured once by photographing one of the wings spread by hand on a reference grid and analysed using ImageJ (National Institutes of Health, USA). Data are presented in table 1. Each flight session lasted a maximum of 20 min, but typically 15 min, since the bird's motivation and willingness was determining how long each flight session could last. The bird was happy to fly through all gaps, but the smallest one was obviously a challenge, so during those flights the bird was a bit more reluctant.

### 2.2. Experimental procedure

The bird was flown along a short corridor that opened up into a room just after the gap (figure 1). Prior to each flight, the bird was sitting on my hand at the beginning of the corridor and the receiving handler was standing on the other side of the gap panel in the room calling and attracting the bird with a treat (millet). The bird would then take off voluntarily and fly from my hand, through the gap and to the receiving handler's hand. The bird was released and received at approximately 1.4 m height which was about the vertical centre of the gap panel. The distance from the take-off location to the gap was approximately 2 m and the distance after the gap to the receiving handler was approximately 1 m. A

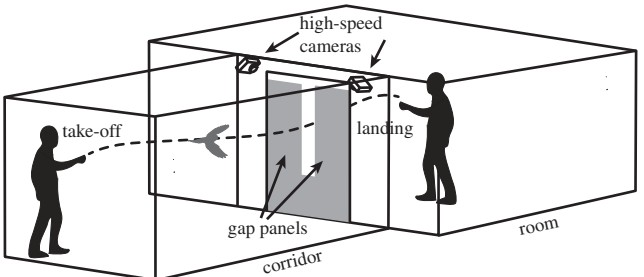

**Figure 1.** Illustration of the set-up. The bird was sitting on the hand of one handler at the beginning of the corridor and took off voluntarily, flew through the gap and landed on the receiving handler's hand in the room on the opposite side of the gap. Two synchronized high-speed cameras were placed above on either side of the gap and recorded the flights of the bird.

**Table 1.** Morphometric data of the bird. The variables included are mass ($M$), wing area ($S$), wingspan ($b$), mean chord ($c$), aspect ratio (AR), wing loading ($Q$) and body width ($B$).

| $M$ (kg) | $S$ (m$^2$) | $b$ (m) | $c$ (m) | AR | $Q$ (N m$^{-2}$) | $B$ (m) |
|---|---|---|---|---|---|---|
| 0.0412 | 0.0123 | 0.29 | 0.042 | 6.9 | 32.9 | 0.036 |

total flight distance of 3 m which was always kept the same for every flight and every gap width in order to assure that all flights were comparable.

We used seven different gap widths: 750 mm (the doorway opening at the end of the corridor and here considered as unrestricted flight), 500 mm, 300 mm (just above the wingspan of the bird), 250 mm (just below the span), 150 mm (approx. 1/2 of span), 100 mm (approx. 1/3 of span) and finally 70 mm (approx. 1/4 of span). The width of the body of the bird, which sets the absolute limit of gap width that the bird theoretically could pass through, was 36 mm (table 1). Which gap width to use was chosen randomly throughout each session to minimize the risk of the bird getting familiar with the sequence of gaps, but each gap width was used for about 3–5 flights in succession, before proceeding to the next random gap width. The gap panels were made out of a plastic foam material (Depron, 6 mm thickness, white) which is very light weight and slightly compliant and they were also hanging from strings allowing them to move which eliminated the risk of the bird getting hurt if colliding with the panels.

## 2.3. Stereo high-speed filming

At the gap, two high-speed cameras (GoPro Hero 4 Black) were installed looking obliquely down on the gap from above, one camera laterally on either side of the gap (figure 1). The cameras were positioned so that the flight up to the gap including the moment of transition was always captured. The cameras were filming at 120 fps with a resolution of 1920 × 1080 pixels and were synchronized using custom syncing electronics ('Bastet' with 'MewPro 2' for the master camera and 'MewPro Cable' for the slave camera. Orangkucing Lab, Tokyo, Japan) that had high temporal precision (less than 5 ns time lag). The master camera controlled the slave and the two cameras were recording continuously throughout each session (which typically allowed for about 15–20 individual flights) and the total recording time was limited by battery life which at this high frame rate was about 20 min maximum. In order to get sharp images with minimum motion blur (particularly from the beating wings), two strong lights were used and the exposure on the cameras were set to −2 EV (exposure value) to increase the shutter speed (reducing exposure time and motion blur).

Cameras were calibrated using the 'stereo camera calibrator' routine in the 'computer vision toolbox' in Matlab. The standard Matlab chequerboard calibration plate was used as calibration target (29 mm chequerboard squares) and filmed at the beginning of each session. The plate was moved in a continuous motion through the volume at various heights and angles in relation to the cameras. Since the GoPro cameras have wide angle field of view with radial fish eye distortion, the calibration was set to calculate intrinsic (as well as extrinsic) camera parameters and to estimate the radial distortion with three coefficients. Individual calibration images that had higher error than 0.8 pixels in the first round was removed, but this was typically only two or three pairs out of the about 55 image pairs

used for each calibration. After removal, the calibration was run again and the mean re-projected pixel error of the final calibrations was about 0.3 pixels. The quality of the calibration was later carefully checked also by manually clicking three corners of the calibration plate at various locations within the volume covered by the cameras. This allowed me to check, at each location, two known lengths of the plate (316 and 258 mm, one perpendicular to the other) and compare this with the result from the three-dimensional calculations (see below). Overall, the error of the distance estimate of the plate at all locations within the volume was 1.3 mm or 0.45%, confirming a high-quality calibration.

## 2.4. Data processing

As described above, the cameras were set to under-exposure by 2 EV in order to reduce motion blur. This resulted in images with very little motion blur and only at the wingtips but a bit dark, so to facilitate digitization, each movie was processed using GoPro Studio (GoPro Inc., USA) to increase exposure and contrast, resulting in images showing the bird clearly (examples shown in electronic supplementary material, videos S1–S9). Important to note is that as a result of this post-processing, the parts of the field of view in the videos that show the opposite side of the gap appear very dark, but in reality, it was well lit and not difficult for the bird to see through the gap. The files were then exported as 'Archive/Edit' which gives the highest quality and retains the full frame rate and resolution. After this, each movie was examined and start and end frames for each flight were noted. Then a custom-written Matlab script extracted each of the individual flights from every full-length movie based on the start and end frames and made individual files for each gap and flight. During this process, every frame was rectified (removing distortion) using the camera parameters from the calibration with the function 'undistortImage' in preparation for calculating three-dimensional positions.

Once pre-processed as above, the, now rectified, individual flight movies were used to manually pinpoint and click the central location on the cere of the bird (the unfeathered skin patch at the base of the upper beak) in every frame for both views. The dark cere is a distinct feature that is easy to see in contrast with the white forehead of the bird. The final calibrated volume after rectifying the images (which crops a bit of the frame) allowed for length of flight trajectories of the bird, used for speed and acceleration measurements, of about 500 mm. For digitization, I used a custom Matlab analysis script 'cliking_gui_two_cams', written by Dr Simon Walker (University of Leeds, UK).

The digitized two-dimensional coordinates ($X$ and $Y$ for both camera views) were then used to calculate three-dimensional coordinates using the function 'triangulate' in Matlab along with the camera parameters from the corresponding calibration.

## 2.5. Calculations of speed and wingbeat frequency

Each flight was analysed using a custom Matlab script. The three-dimensional coordinates were smoothed by fitting a function using 'csaps' (smoothing parameter $1–10^{-7}$) for each axis to reduce potential digitization and calibration errors but primarily to generate a continuous function. To calculate the flight speed of the bird at each time step, the derivative of the fitted csaps-function was calculated using 'fnder' and the total speed as $V_{\text{Tot}} = \sqrt{V_x^2 + V_y^2 + V_z^2}$, horizontal speed as $V_H = \sqrt{V_x^2 + V_y^2}$ and vertical speed just as the $V_z$ component. Overall flight speeds for each flight were calculated as the average of the speeds over all time steps up to the gap.

Wingbeat frequency (WBF) was calculated by counting the number of wingbeats ($N_{\text{wb}}$) before arriving at the gap (about 3–6 depending on the flight speed of the bird) and noting the frame number at the start of the first wingbeat ($Fr_s$) and the frame number at the end of the last ($Fr_e$), along with the frame rate of 120 fps wingbeat frequency was calculated as WBF $= N_{\text{wb}}/(1/120)(Fr_e - Fr_s)$.

## 2.6. Calculations of accelerations and decelerations

To get the average accelerations and decelerations of the bird during the flight towards the gap, I fitted a linear line using 'polyfit' of first order in Matlab to the velocity data calculated as described above. The slope of the line then represents the mean acceleration/deceleration during the approach.

## 2.7. Measuring wingtip distance during gap transition

To measure how close together the bird kept its wingtips when flying through the gaps smaller than the wingspan, I digitized the two wingtips at the critical instance just at the gap opening for sequences that

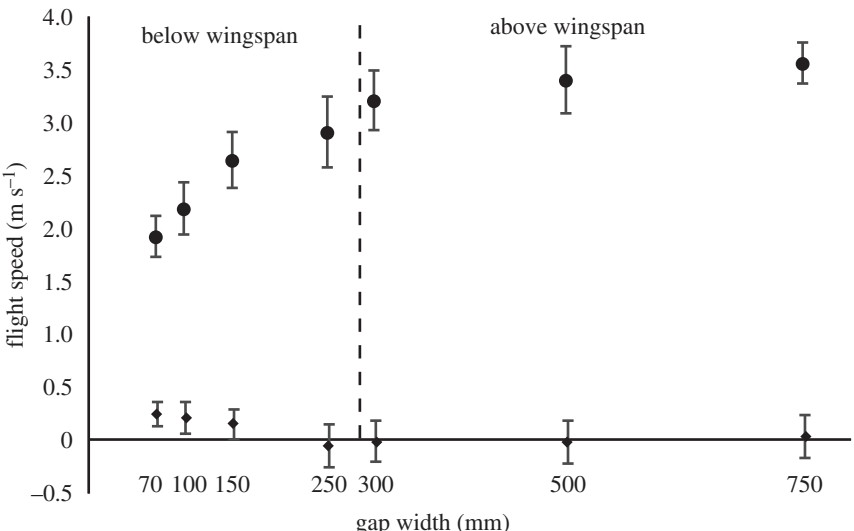

**Figure 2.** Flight speeds of the bird flying through the different gap widths. Circles show horizontal speed and diamonds show vertical speed. The dashed vertical line shows the wingspan of the bird. Error bars represent ±1 s.d.

allowed for it (sometimes the wingtips were hidden behind the body in one of the camera views), calculated the three-dimensional positions using the same procedure as described in §2.3, and calculated the distance between the wingtips. The number of sequences that were possible to use for this analysis was 5, 12, 11 and 11 for gaps 70, 100, 150 and 250 mm, respectively.

## 3. Results

We recorded in total 139 individual flights across all gap widths. We recorded at least 20 flights for every gap apart from the smallest (70 mm) where we were able to get 14 flights, simply due to the fact that it was a challenge for the bird, which made it a bit more reluctant to fly through it. It only happened five times over all flights that the bird gently bumped into one of the edges of the panel opening (and only for the 70 and 100 mm gaps) during the transition through the gap (a mere 3.6%) and even in these cases, the bird flew through the gap and continued on the other side without any obvious interruption.

### 3.1. Influence of gap width on flight speed

At the widest opening (the width of the doorway, 750 mm), the bird flew at an average horizontal speed of 3.6 m s$^{-1}$ which we can here, for comparisons, see as a reference speed for what the bird chose to fly at over the flight distance used in the study, if unrestricted. For the gaps wider than the wingspan (300, 500, 750 mm), the average horizontal speed only decreased slightly with decreasing gap width (from 3.6 to 3.2 m s$^{-1}$), although significantly (ANOVA, $F_{2,59} = 8.74$, $p = 0.0005$; figure 2). When gap width became shorter than wingspan, the speed decreases at a distinctly faster rate (from 2.9 m s$^{-1}$ at 250 mm gap to 1.9 m s$^{-1}$ at 70 mm gap). The difference is highly significant between these four gaps (ANOVA, $F_{3,73} = 46.86$, $p < 0.001$; figure 2).

The vertical speed of the bird was close to zero for all gaps wider than the wingspan (300, 500 and 700 mm) and unchanged between them (ANOVA, $F_{2,59} = 0.39$, $p = 0.68$; figure 2) which means the bird flew in level flight up to the gap. With gap widths smaller than the wingspan, the bird had a slight upward vertical speed that changed between the gaps (ANOVA, $F_{3,73} = 13.17$, $p < 0.001$; figure 2).

### 3.2. Influence of gap width on wingbeat frequency

Over the whole range of gap widths, the bird altered its wingbeat frequency (ANOVA, $F_{6,133} = 15.50$, $p < 0.001$; figure 3). However, the response was not following a continuous change across widths, but instead, there was a distinct change in wingbeat frequency from the four widest gaps to the three smallest. Within the four widest gaps, the wingbeat frequency was on average 15.5 Hz and

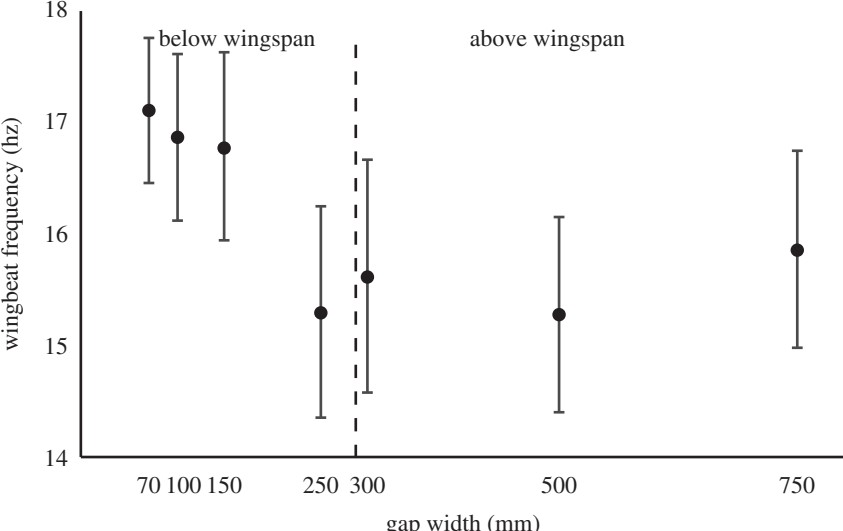

**Figure 3.** Wing beat frequency of the bird across the different gap widths. The dashed vertical line shows the wingspan of the bird. Error bars represent ±1 s.d.

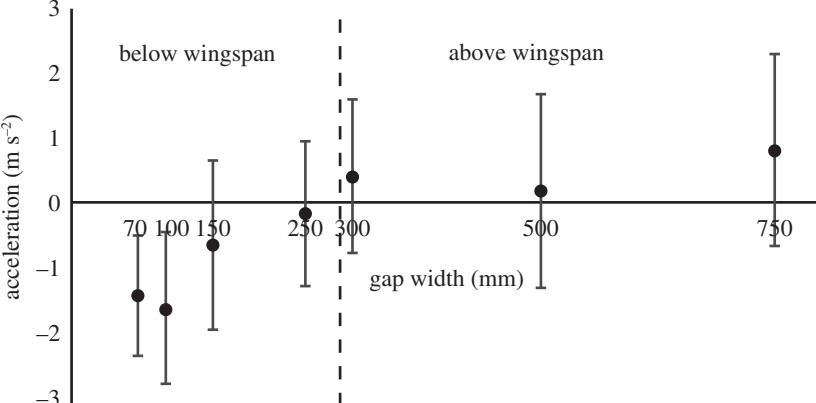

**Figure 4.** The bird performing decelerations or accelerations for different gap widths. For gaps above wingspan (greater than 290 mm), the bird kept a more or less a steady flight speed or accelerated slightly. It was not until gap width was below wingspan (less than 290 mm) that the bird was on average decelerating (and increasingly so with gap tightness). Error bars represent ±1 s.d.

unchanged (ANOVA, $F_{3,78} = 1.8$, $p = 0.15$) and for the three smallest gaps, it was on average 17 Hz and unchanged (ANOVA, $F_{2,55} = 0.88$, $p = 0.42$; figure 3).

## 3.3. Braking behaviour

When flying through gaps larger than the wingspan, the bird kept the same flight speed or was on average accelerating slightly. There was no difference in the small accelerations between these three gaps (ANOVA, $F_{2,59} = 1,12$, $p = 0.33$; figure 4). When the gap width was smaller than the wingspan, the bird was on average decelerating (negative acceleration). Deceleration, or braking, was more pronounced as gap width got smaller and there was a highly significant difference in decelerations between these four gaps (ANOVA, $F_{3,73} = 6.65$, $p < 0.001$; figure 4).

If looking at the mean change of flight speed of the bird just before the gap (here chosen as the last 0.2 s before arriving at the gap) for the different gap widths, we see that the bird is not only flying slower it is also braking with higher decelerations, resulting in a potential twofold cost (figure 5). We can see this from the progressively increasing negative slope of the lines in figure 5 and from the ANOVA test of the mean acceleration (which is the same as these slopes) as presented above. For the sake of clarity in figure 5, only the average lines are plotted, but means and standard deviations of the coefficients are

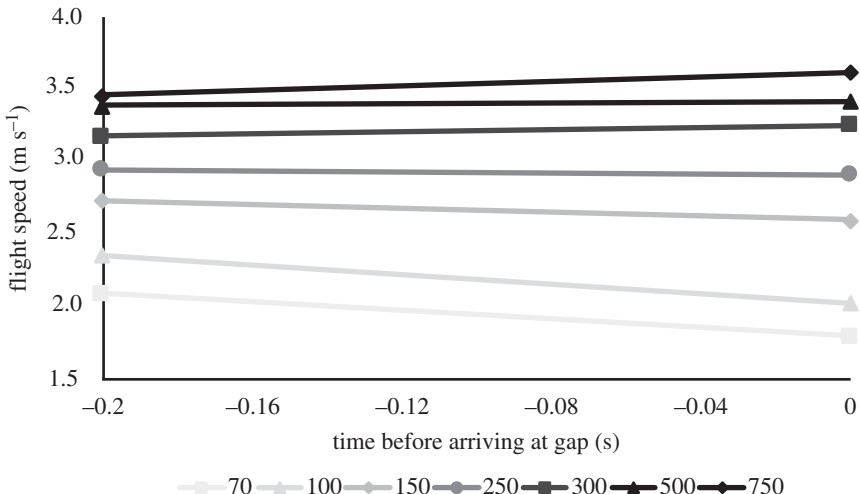

**Figure 5.** Speed change during the last 0.2 s before the gap for the different gap widths. As seen previously in figure 2, the overall flight speed decreases with decreasing gap width. At the gaps below wingspan (less than 290 mm), the bird is decelerating, while for gaps above (greater than 290 mm), it is either steady or slightly accelerating. This figure shows both the slope (as in figure 4) and the overall flight speed for each gap (which figure 4 does not). For the sake of simplicity and clarity of the plot, the lines only show the mean slopes and offsets, but the standard deviations for both coefficients are presented in electronic supplementary material, table S1.

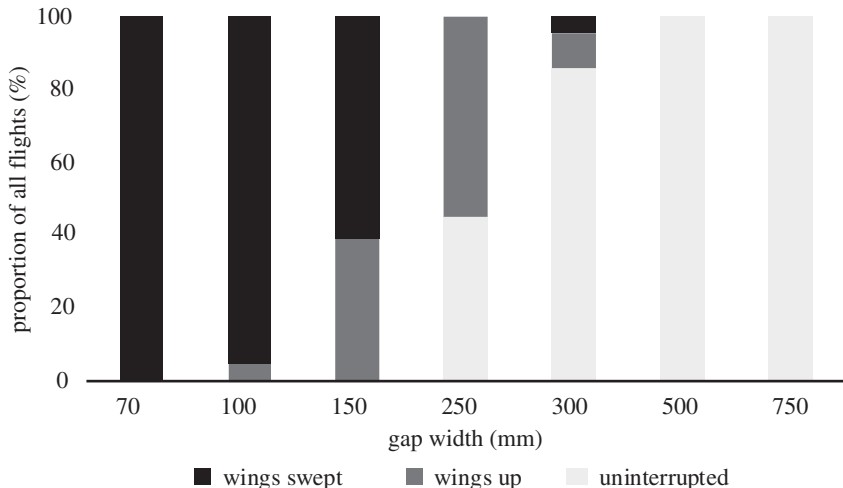

**Figure 6.** Postures adopted by the bird when flying through the different gap widths. For the two wider gaps, the bird did no change (uninterrupted). At the gap widths close to the wingspan, the bird started to adopt two main postures; wings paused at the top of the wingstroke (wings up) or wings swept backwards at mid-stroke (wings swept). At the two smallest gap widths, the swept wings were dominating.

presented in electronic supplementary material, table S1. The standard deviations of the slopes can also be visually interpreted from the error bars of figure 4 and standard deviations of the offset can be assessed from the error bars in figure 2.

## 3.4. Postures when flying through the gap

For the two widest gaps (750 and 500 mm), that were clearly wider than the wingspan, the bird did not interrupt its wingbeat (figure 6; electronic supplementary material, videos S1 and S2). When the gap width was close to the wingspan (300 mm), the bird most of the time did not interrupt its wingbeat (electronic supplementary material, video S3), but in a few cases started to adopt two different wing postures; wings paused at the top of the upstroke or wings swept backwards at mid-stroke (figure 6; electronic supplementary material, figure S1*A* and *B*). At just below wingspan at 250 mm gap, the bird

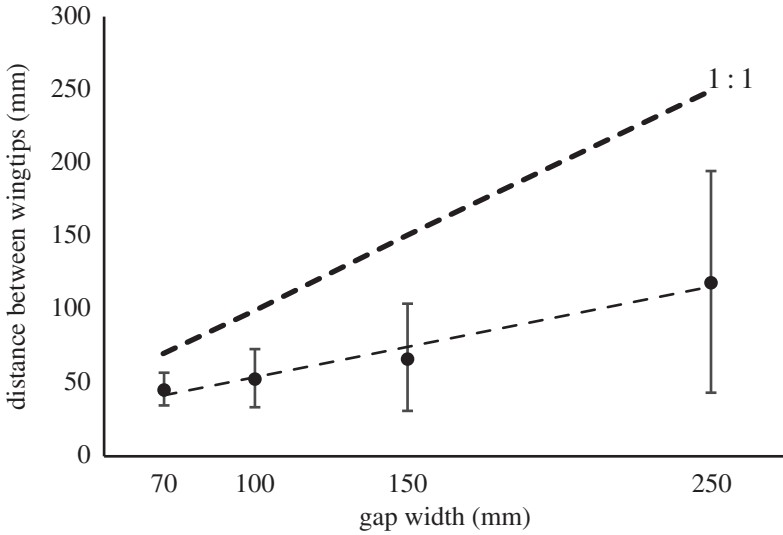

**Figure 7.** Distance between wingtip at the critical instance of traversing the gap in relation to the gap width. A line parallel to the 1 : 1 line would indicate a constant margin across gaps, while a diverging line like this one tells us that the larger the gap, the larger the margin.

either timed the downstroke to fit without interrupting the wingbeat (electronic supplementary material, video S4, figure S1C) or used the wings-up posture (electronic supplementary material, video S5). When gap width was about half wingspan (150 mm), the bird always interrupted its wingbeat and used either of the two postures (electronic supplementary material, videos S6 and S7). At the 100 mm gap, the swept wings posture was dominating and at the smallest gap width of 70 mm, the bird exclusively used the swept wings posture (figure 6; electronic supplementary material, videos S8 and S9).

In a few cases, for the gaps smaller than the wingspan, the bird rolled as it was going through the gap, in total 8% of the flights. Of these flights, 5.8% was a roll to the left and 2.2% was a roll to the right. In only one instance, at gap 250 mm, did the bird use only the roll as a means to get through the gap. In all other flights, the roll was in combination with one of the postures.

For the gap width lower than the wingspan, when the bird was adopting one of the postures, it would reduce the distance between the two wingtips in order to fit through the gap. The relative distances between the wings in relation to the gap width were not constant (line diverging from 1 : 1 line in figure 7), so the margins were different ('margins' here and forthwith refers to the difference between the gap width and the wingtip separation when transitioning the gap). The smallest gap width, gives us a proof of what the bird is at least capable of in terms of small margins (average 24 mm), but this is not the margin that the bird chooses to use for the larger gaps. With increasing gap width, the bird typically used larger margins (figure 7). The average wingtip distances for the four gaps smaller than wingspan were 46, 53, 70 and 119 mm for the 70, 100, 150 and 250 mm gaps, respectively. This gave average total margins of 24, 47, 80 and 131 mm for the four gaps.

## 4. Discussion

### 4.1. The challenge

The aim for this study has been to investigate how a bird deals with the problem of flying through tight gaps and what the costs that come with it are in terms of flight speed and kinematic changes. The implications of this can be viewed in the perspective of birds flying in complex environments in the wild. In a forest or bush environment, the birds need to negotiate through various gap widths constantly along their flight path when flying. The challenge that the budgerigar in this study was faced with is probably a relatively easy one since it only involves traversing a single gap, whereas in nature after one gap, there will soon be another one to deal with. Having said that, the results from this study still give us a good idea of how a bird solves the problem and in what ways its flight performance is impaired.

## 4.2. Cost of flying through gaps

### 4.2.1. Altered flight speed

One of the most obvious results in this study is the reduction of the overall flight speed that the budgerigar chose to fly at when approaching the smaller gaps. For gaps above the wingspan, there was only a small difference in horizontal flight speed, but for the gaps smaller than the wingspan, the reduction in flight speed was clear (figure 2). This first of all tells us that the bird was able to early on assess the gap width and that it adjusted its flight speed accordingly. Regarding the vertical flight speed, for gaps wider than the span, the bird had a vertical speed close to zero (level flight), but when the gaps got narrower than the span, the bird had a slight vertical upward flight speed of about $0.2 \, \mathrm{m\,s^{-1}}$ (figure 2). This also indicates that the bird has the ability to assess the gap width rather accurately and although we cannot know for sure the reason for the bird to gain altitude before the gap, it is perhaps likely that it is to compensate for the height loss that will happen from the pause in the wingbeat that the bird has to do to get past the narrow gaps. This awareness and ability that the birds possess, to assess the gap width in relation to its own wingspan, was also shown by Schiffner *et al.* [3], Bhagavatula *et al.* [5] and Williams & Biewener [14]. When including these narrow gap widths as I have done in this study, down to 1/4 of the wingspan, we can see that speed changes continuously over the range of gap widths, albeit nonlinearly. This is different from what was reported by Vo *et al.* [4], where the budgerigars were always approaching the gaps at about $4 \, \mathrm{m\,s^{-1}}$ regardless of the gap width. Vo *et al.* [4] proposed that the reason for this was that by doing so, the birds could keep the same distance or time to the gap as trigger for when to perform wing closure. However, the range of gap widths used in that study (240–380 mm) were chosen to bracket the wingspan of the birds and the challenge for the birds may not have been severe enough for them to respond by reducing their flight speed. Worth to note is that they did include one 130 mm gap width as a case where all birds regardless of wingspan would be challenged and there they indeed found a reduction in flight speed. The speed they measured is similar to the speed at 150 mm gap of my bird (just above $2.5 \, \mathrm{m\,s^{-1}}$). In this current study, there is very little impact on flight speed above gap width 250 mm, but we do see a clear response that follows gradually when going further down in gap width, suggesting that the bird does adjust its flight speed to the current gap widths rather carefully. The reason for the bird to lower its flight speed with tightness of the gap could be a way to minimize the risk of collision and also to minimize the impact of collision if it happens.

The vertical speed after the gap was not possible to measure in this study, but there will most likely be a slight loss of altitude from the pause in the wingbeat cycle during the gap transition, so we can view the altitude gain from the upward vertical speed before the gap as an indication of what a bird in the wild needs to do before (or after) traversing a tight gap.

The bird lowered its average horizontal flight speed from $3.6 \, \mathrm{m\,s^{-1}}$ at the unrestricted opening (similar to [4]) to $1.9 \, \mathrm{m\,s^{-1}}$ at the smallest gap, which is almost a 50% reduction (figure 2). If we put this in the perspective of a bird in the wild and assume a similar response, this is a clear cost. First of all, it means that the overall duration of flight for getting from one place to another increases in direct proportion, so if speed is reduced by half the flight duration naturally is doubled. Since flight is energetically very costly and only an economical mode of locomotion at favourable fast flight speeds (e.g. [15]), this is obviously affecting the energy budget of the bird. This is the simplest way to look at this cost. If we also consider that the power to fly typically does not scale linearly with flight speed, but rather follows a U-shaped curve, with higher power required to fly both slower and faster than a certain minimum power speed (e.g. [16–21]) or L-shaped with elevated power consumption for the slow range [22], then we can understand that having to reduce flight speed most likely comes with more cost than just prolonged flight duration. This is a general argument for all flying animals, but considering budgerigars in particular the muscle recruitment has indeed been shown to be consistent with meeting flight power requirements that vary in a U-shaped pattern with speed [23,24] as does the respiratory rate [7]. It means that when a bird (or any flying animal) is forced to fly at a lower-than-preferred flight speed, it is using disproportionally more power to fly and this adds further to the cost. The increased cost of flight means that either more energy needs to be allocated to flying or the flight distance has to be reduced.

### 4.2.2. Altered wingbeat frequency

When the bird was flying through gaps wider than the wingspan, it had an average wingbeat frequency of 15.5 Hz. Once the gaps got clearly smaller than the wingspan, the bird increased its wingbeat frequency distinctly, to on average 17 Hz (figure 4), indicating that it is operating at a suboptimal

flight speed which forces it to alter its kinematics to be able to fly at the slow flight speed. This is part of the reason that the power consumption is elevated at slow flight speeds. It is, in this case, a 10% increase in wingbeat frequency and the simplest relationship would be that the increase in energy cost scales linearly so that a 10% increase in wingbeat frequency results in a 10% increase in energy consumption.

So, not only is the bird flying slower which means it takes longer to get to the destination, most likely with elevated power consumption, it is also beating the wings faster. All in all, traversing small gaps comes with multiple costs to the flying bird.

## 4.3. Braking behaviour

When the bird was flying through the gaps that were wider than the wingspan, it was either more or less keeping a steady flight speed or accelerating only slightly. This is a good indication that the bird had time to get up to (or at least close to) a preferred speed over the flight distance used in the experiments. When the gaps were smaller than the wingspan, the bird was on average decelerating, braking. A part of this deceleration may come from the slight vertical flight speed that the bird has before the narrow gaps (trading kinetic energy against potential), but since that speed is rather small, the majority of the deceleration is probably active braking. Regardless, the bird is decelerating and we know already that the bird had lower flight speeds at narrower gap widths to begin with (figure 2), and then we also see that the bird is braking more with reducing gap widths (figure 5). This means that the bird not only flies slower on average for the smaller gaps, it also brakes from this slow flight speeds at a higher rate in order to safely and successfully traverse the gap. This results in the speed at which the bird needs to accelerate from after the gap being even lower than the average speed. If we consider again a bird flying in the wild which may be flying through multiple gaps along its flight route, it has to, not only, reduce the flight speed (momentarily pushing it further away from the preferred speed), it also has to accelerate after every gap, which will inevitable involve a cost as well.

## 4.4. Wing postures

When flying through the gaps wider than the wingspan, the bird would pass through the gap without interrupting the wingbeat. When the gap width was smaller than the wingspan, the bird used two solutions, either pausing at the end of the upstroke or pausing at mid-stroke and sweeping the wings back (sometimes adopting the posture during upstroke, sometimes during downstroke depending on the timing of the wingbeat phase when arriving at the gap). The upstroke pause was dominating for the gap width close to the wingspan while swept wings was dominating for the smaller gaps. This is consistent with the previous findings for budgerigars [3] and also for pigeons that adopt the same two postures [14].

Interesting to note is that the bird tended to, for the two smallest gaps, time the final wingbeat just before the gap, sometimes even brushing gently up against the panels with the wingtip feathers, before sweeping the wings back and 'slide' through the gap (electronic supplementary material, videos S8 and S9). For gap 150 mm, the final wingbeat was timed a bit earlier before the gap (electronic supplementary material, videos S6 and S7). At first, this is perhaps surprising since you assume that the smaller gaps are more difficult to traverse, but this is most likely an effect of the overall higher flight speed at the wider gap which means the bird needs a longer distance in order to have similar time to adopt the posture. For the critical gap width of 250 mm (just below wingspan), the bird would either time the wingstroke so that the wings were towards the end of downstroke just at the gap opening allowing the bird to pass through without pausing (electronic supplementary material, video S4), or (if timing was off) the bird would pause at the end of upstroke and glide through the gap (electronic supplementary material, video S5). In a couple of flights through gap 250 mm, the bird would pause at the end of upstroke a bit earlier than usual and then time a wingstroke just at the gap as described above, giving the impression that it was actively timing a downstroke to still fit through the gap if the timing was not too off. Obviously, these events only represent a few flights, so caution is advised for drawing any general conclusions from this, but they may be viewed as examples of some of the solutions that a bird may use.

Similar to what Schiffner *et al.* found [3], it is clear from these results that the bird avoids interrupting the wingbeat as far as possible and when interruption is unavoidable, it keeps it as brief as possible. This makes perfect sense if the bird is minimizing the detrimental effects on flight performance. In a recent study [12], budgerigars were first allowed to over a period of time establish a preferred flight path from one perch to another inside a tunnel and then an obstacle was introduced in the middle of that

flight path. The birds would only deviate from their established path just before reaching the obstacle and veer over it as quickly as possible to return to the preferred path [12]. This is yet another indication that the birds are minimizing the changes they have to do.

Similar behaviour as seen in my budgerigar has been reported for pigeons when they are challenged with various gap widths [14]. When flying through the wider openings that were easy challenges for the pigeons (but still smaller than the wingspan), they adopted the posture of pausing at the end of upstroke. Only for gaps that were a real challenge for the birds did they adopt the swept wing posture. In that paper, the authors propose that the birds are essentially trading speed for stability when traversing smaller gaps. They concluded that the swept wing posture is more stable than the paused wing posture, while it is less efficient [14]. The behaviour of the budgerigar in this study is very similar to that of the pigeons, suggesting some generality in this behaviour. Another potential reason for the bird to prefer to pause at upstroke over the swept wings posture when possible is that it may be easier for the bird to resume the wingbeat cycle after passing the gap than from the swept wing posture.

## 4.5. Safety margins

When the bird was flying through the gaps smaller than the wingspan and adopting one of the postures, it would bring the wingtips together in order to fit through the gap. For the two smallest gaps, the wings would often (approx. 2/3 of the flights) even cross over the body. At the smallest gap width of 70 mm, the bird had an average wingtip distance of 46 mm which gives it a 24 mm total safety margin, or a mere 12 mm on either side (figure 7). This gives us a proof of what the bird is at least capable of in terms of using small safety margins, so in theory the bird could use this margin for all gaps smaller than the wingspan and that way it would minimize the changes to its wingbeat. However, this is not how the bird chooses to do it. Instead, when that gap width gets larger, the bird allows for a larger safety margin on average, as the divergence between the fitted line and the 1:1 line in figure 7 shows us. If the margin had been the same for all gaps, the two lines would have been parallel. It means that we see both what the bird *can* do when forced and what it *prefers* to do if able to choose. This result indicates that the bird has a strategy for how to safely traverse the gaps, but to some degree, this probably also reflects a larger need for safety margins at the larger gaps because of the higher flight speed. Another potential reason may be that in free unrestricted flight, the bird may have a preference for an optimum, energy-efficient wingstroke amplitude and this may dictate the preferred wingtip separation at the top of the upstroke. The use of tight safety margins while flying through gaps narrower than the wingspan may be a consequence of attempting to maintain the minimum wingtip separation as close to the optimum value as possible.

The variation in the span that the bird adopted increased with increasing gap width (figure 7), which follows logically with that the bird is freer to choose the wing posture and the margin for the larger gap widths. So, even if the average span that the bird adopted at, for example, 250 mm allows for a large safety margin compared to what the bird has proven to be capable of, the range of the spans is large and if we look instead at the maximum span that the bird used it was 238 mm, 12 mm total margin, 6 mm on either side. This type of large span and small margin at the 250 mm gap happened in the cases where the bird did not interrupt its wingbeat but timed the downstroke to just fit inside the gap as described above. The maximum span measured for the smallest gap of 70 mm was 58 mm, which also gives a total margin of 12 mm. This shows that the bird is capable of operating with very small safety margins, when it needs or wants to.

## 4.6. Concluding remarks

In this study, a budgerigar shows us how it deals with the challenge of flying through gaps of various widths. We see that as long as the gap is wider than the wingspan, the bird flies with the same flight speed, but when the gap width is lower than the span, horizontal speed is reduced with reducing gap width while a small upward vertical speed results in a slight gain in altitude before the gap. The bird also increases its wingbeat frequency when the gap width is smaller than the wingspan. I conclude that this results in several costs for a bird flying in complex environments in the wild, of which the main costs may be: (i) the reduced flight speed first of all increases the flight duration and hence the energy consumption to get from point $A$ to $B$, (ii) due to the underlying U-shaped power curve of flying animals, there is an added cost associated with having to reduce the flight speed from the preferred, and (iii) with an elevated wingbeat frequency on top of this, a third direct cost is included. The bird further shows us that for gaps smaller than the wingspan, it brakes before the gap and this

braking behaviour gets more pronounced with the tightness of the gap. It is capable of flying through a gap with a small margin as shown for the smallest gap width, but when flying through the wider gaps, it chooses to on average allow for a larger safety margin.

The results of this study give us a good idea about the types of cost associated with flying through gaps as well as teaching us what is a valid solution to the problem in terms of flight speeds, braking behaviour, wing postures and safety margins.

Ethics. The bird used for these experiments is our own pet that was kept throughout the experiments and during them in our own home and was never exposed to any suffering or stress. Swedish Board of Agriculture states in SJVF 2019:9, L150, Chapter 2, 18 §, that a privately owned pet that is kept in its usual housing facility during experiments and is not euthanized, put through any invasive treatments (such as surgical or blood sampling) or exposed to any suffering, can be used for mild experiments without specific permission. The bird did all flights voluntarily and it was only encouraged with positive reinforcement through treats and verbal reward and never using negative reinforcement through punishment.

Data accessibility. Data, Matlab code and Excel result files are available from the Dryad Digital Repository: https://doi.org/10.5061/dryad.rr4xgxd8j [25].

Competing interests. I declare I have no competing interests.

Funding. This research was funded by a grant from the Swedish Research Council (VR.se) to P.H. (2018-04292).

Acknowledgements. First of all, I thank our beloved little budgie, 'Poppen', for without complaining performing the impressive flights in exchange for millets. I also thank my wife, Teresa Kullberg, and our daughter, Alice Henningsson, for invaluable help during experiments and for inspiring conversations and discussions about the flight of the budgerigar and the results along the way. Without their help, the study would not have been possible to perform.

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
