## [Peer Review File · Royal Society Open Science]

Review History

Decision letter (RSOS-211072.R0)

Dear Dr Henningsson:

I am pleased to inform you that your manuscript entitled "Flying through gaps – How does a bird deal with the problem and what costs are there?" is now accepted for publication in Royal Society Open Science.

If you have not already done so, please remember to make any data sets or code libraries 'live' prior to publication, and update any links as needed when you receive a proof to check - for

instance, from a private 'for review' URL to a publicly accessible 'for publication' URL. It is good practice to also add data sets, code and other digital materials to your reference list.

on behalf of Professor Brooke Flammang (Associate Editor) and Professor Kevin Padian (Subject Editor).

Subject Editor Comments to the Authors (Professor Kevin Padian):
Comments to the Authors:

Thanks for your revisions of the manuscript. As one reviewer says, doing this work at home with one pet bird is likely the result of the Covid-lockdown, and as a preliminary study with a simplified methodological approach it seems to be a good start. I agree that more animals eventually could be used, but I see this as a pilot study that does add something to our knowledge and to our methodological approaches (in an unusual way). In my view you've responded very constructively to the comments, and so I would like to accept it. The objections raised are good ones, but they can be addressed in future work, once the community assesses it. It will be interesting to see how it's received -- hopefully for what it is rather than for what it isn't.
